# Titration of Androgen Signaling: How Basic Studies Have Informed Clinical Trials Using High-Dose Testosterone Therapy in Castrate-Resistant Prostate Cancer

**DOI:** 10.3390/life11090884

**Published:** 2021-08-27

**Authors:** Steven K. Nordeen, Lih-Jen Su, Gregory A. Osborne, Perry M. Hayman, David J. Orlicky, Veronica M. Wessells, Adrie van Bokhoven, Thomas W. Flaig

**Affiliations:** 1Department of Pathology, University of Colorado Denver Anschutz Medical Campus, Aurora, CO 80045, USA; steve.nordeen@cuanschutz.edu (S.K.N.); david.orlicky@cuanschutz.edu (D.J.O.); adrie.vanbokhoven@cuanschutz.edu (A.v.B.); 2Division of Medical Oncology, Department of Medicine, University of Colorado Denver Anschutz Medical Campus, Aurora, CO 80045, USA; Lih-Jen.Su@cuanschutz.edu (L.-J.S.); gregory.osborne@cuanschutz.edu (G.A.O.); Perry.Hayman@colorado.edu (P.M.H.); veronica.wessells@cuanschutz.edu (V.M.W.)

**Keywords:** prostate cancer, castrate-resistant prostate cancer, androgens, testosterone, androgen receptor, androgen signaling, bipolar androgen therapy, high testosterone resistance, androgen-independent prostate cancer, clinical trials, human, cancer therapy

## Abstract

Since the Nobel Prize-winning work of Huggins, androgen ablation has been a mainstay for treatment of recurrent prostate cancer. While initially effective for most patients, prostate cancers inevitably develop the ability to survive, grow, and metastasize further, despite ongoing androgen suppression. Here, we briefly review key preclinical studies over decades and include illustrative examples from our own laboratories that suggest prostate cancer cells titrate androgen signaling to optimize growth. Such laboratory-based studies argue that adaptations that allow growth in a low-androgen environment render prostate cancer sensitive to restoration of androgens, especially at supraphysiologic doses. Based on preclinical data as well as clinical observations, trials employing high-dose testosterone (HDT) therapy have now been conducted. These trials suggest a clinical benefit in cancer response and quality of life in a subset of castration-resistant prostate cancer patients. Laboratory studies also suggest that HDT may yet be optimized further to improve efficacy or durability of response. However, laboratory observations suggest that the cancer will inevitably adapt to HDT, and, as with prior androgen deprivation, disease progression follows. Nonetheless, the adaptations made to render tumors resistant to hormonal manipulations may reveal vulnerabilities that can be exploited to prolong survival and provide other clinical benefits.

## 1. Introduction

Prostate cancer is the most common non-skin cancer in men. While most prostate cancers are relatively indolent and are cured by local treatment, recurrent cancers and cancers that are advanced at diagnosis are responsible for over 30,000 deaths annually in the United States alone. In this work, we briefly address the current status of clinical management of recurrent or advanced prostate cancer. Ever more efficacious suppression of androgen signaling has been the central focus of therapeutic development for decades. However, progress in the treatment of men whose disease has progressed in the face of continued androgen suppression has been limited. Recent trials employing the approach of high-dose testosterone (HDT) therapy in castrate-resistant prostate cancer (CRPC) show clinical benefit and improved quality of life in a subset of CRPC patients who otherwise have few treatment options, all with potential for significant side effects. We review preclinical studies that supplied support for the heretical (to some) concept of administering testosterone to prostate cancer patients. Using previously unpublished data from our laboratories, we illustrate the types of findings that have led to the reassessment of the potential clinical utility of manipulation of the androgen environment in prostate cancer. Finally, we discuss the implication of such findings on the future of HDT and how this approach may be optimized further.

### 1.1. Advanced/Recurrent Prostate Cancer Standard of Care: Advancements and Shortcomings

Targeting of the androgen receptor (AR) axis has been the foundation of therapy for advanced prostate cancer for over 75 years. The seminal work by Huggins and Hodges in the 1940s led to the widespread use of surgical orchiectomy for symptomatic metastatic prostate cancer [1]. Medical castration was introduced in the 1980s and has largely supplanted surgical orchiectomy as the preferred method of androgen-deprivation therapy (ADT) [2]. Other recent advances have further increased our ability to inhibit the AR axis: abiraterone is an inhibitor of CYP17 (cytochrome P450 17A1, or 17α-monooxygenase, 17α-hydroxylase, 17,20-lyase) that effectively blocks adrenal production of testosterone [3], and enzalutamide represents a new class of potent and specific antiandrogens [4]. Both agents were initially approved in the metastatic, castration-resistant setting, but are now used for hormone-sensitive disease with significant prolongation in the overall survival. The combination of abiraterone or enzalutamide with standard ADT provides a very effective blockade of the AR axis. Arguably, further clinical benefit from more effective AR inhibition seems unlikely.

The use of more-intensive strategies to block androgen signaling has now been extended into the metastatic, hormone-sensitive setting. For example, in the LATITUDE trial, men with metastatic, castration-sensitive prostate cancer were randomized to ADT with or without abiraterone as initial systemic therapy [5]. Patients included in this study had one of three high risk factors: Gleason score of ≥8, presence of three or more lesions on bone scan, or presence of visceral organ involvement. With a median follow-up of 51.8 months, the overall survival was improved with abiraterone treatment (median 53.3 months), compared to the placebo group (36.5 months), with a hazard ratio of 0.66 (95% CI 0.56−0.78; *p* < 0.0001). In the combination group, a significant long-term benefit was realized, with the median time to subsequent prostate therapy and time to secondary progression-free survival in excess of 4 years.

Despite more aggressive upfront therapy, the outcomes of metastatic prostate cancer remain poor, especially after the development of castrate-resistant disease. For example, in later-stage patients with castration-resistant prostate cancer who have progressed after docetaxel chemotherapy and an androgen-signaling-targeted inhibitor (e.g., abiraterone or enzalutamide), the use of secondary chemotherapy versus an additional androgen-signaling-targeted inhibitor has been studied. In this setting, the use of secondary chemotherapy was superior to an additional androgen-signaling-targeted inhibitor; however, the median overall survival was 13.6 months with cabazitaxel versus 11.0 months with the androgen-signaling-targeted inhibitor, which highlighted the limits of treatment efficacy in later-stage prostate cancer patients.

While both preclinical and human studies have identified potential mechanisms for resistance to ADT, such as expression of AR variants [6] and overexpression of c-myc [7,8], currently there are no effective strategies to target these mechanisms Thus, there is an unmet need for effective therapies for men progressing on androgen-targeting therapy.

### 1.2. Friend and Foe: Rethinking the Complex Role of Androgen Signaling in Prostate Cancer

The centrality of androgen signaling in prostate cancer was highlighted by the report of Chen et al. in that a moderate increase in AR gene expression was the only consistent change in gene expression in CRPC, and that this increase in receptor mRNA and protein was required for resistance [9]. There are longstanding observations in preclinical models of CRPC that androgens can have a suppressive effect [8,10,11,12,13,14,15,16,17,18,19]. While periodic suspension of androgen suppression showed promise in animal models, in human trials intermittent androgen suppression did not produce the desired benefit in tumor progression. The most important clinical trial in this area, SWOG9346, was designed as a noninferiority trial of continuous versus intermittent ADT in untreated metastatic patients [20]. The results of this large trial were inconclusive with respect to survival between the treatment arms, and decreased survival with intermittent ADT could not be ruled out. There were initial improvements in quality of life in those treated with intermittent ADT, including with erectile function, but these largely dissipated beyond the first 9 months of starting the trial. Importantly, animal studies differed from the human trials in that exogenous testosterone was administered in the animal studies, whereas in humans, testosterone was allowed to recover simply by suspending the chemical suppression of androgens. In this older population of men, reestablishment of physiologic testosterone levels can be relatively slow, more than a year in some patients, and incomplete. This may allow tumor cells sufficient time to adapt to the changing androgen environment in contrast to the animal studies where treatment protocols permit rapid alteration of androgen levels. However, the translation of the preclinical findings to the clinical setting has been delayed in large measure due to concerns about the potential stimulatory effect of androgen in advanced prostate cancer patients. Indeed, in the landmark paper by Huggins and Hodges detailing the benefit of surgical castration, the effect of androgen administration was also reported in a small number of prostate cancer patients, establishing the long-held clinical concern about this approach: “Testosterone propionate caused an increase of serum acid phosphatase (tumor marker of that era)] above the pre-injection level in these patients. Following the cessation of injections, there was a decline to the preliminary level.” [1].

More recently, there has been renewed clinical interest in the potential therapeutic and quality-of-life benefit of HDT in CRPC. Much of the recent work has been spearheaded by Sam Denmeade and colleagues at Johns Hopkins University, including a small pilot clinical study in patients with CRPC published in 2015. In this study, investigators administered HDT as bipolar androgen therapy (BAT), in which patients were given a high-dose injection of testosterone cypionate monthly [21]. This produced transient supraphysiologic testosterone levels of more than 1500 ng/dl in some patients. Although etoposide was also used in the early studies, subsequent clinical trials of BAT have not utilized etoposide due to its unclear contribution to the clinical effect noted. In this pilot, prostate-specific antigen (PSA) reductions were seen in 7 of 14 evaluable patients, with radiographic responses also noted in 5 of 10 evaluable patients with radiographically measurable disease. Of the 7 patients with a PSA response, the median time to PSA progression was 221 days. Subsequent studies of BAT have been performed in CRPC patients progressing after treatment with enzalutamide, with 9 of 30 (30%) demonstrating a PSA response [22]. Building on this initial experience, a multicenter randomized trial of BAT has now been completed [23]. In this trial, men with metastatic CRPC previously treated with abiraterone, but without previous potent next-generation antiandrogen (e.g., enzalutamide) exposure, were randomized to enzalutamide versus high-dose intramuscular testosterone with crossover at progression. A PSA decline of 50% was noted in 28% with BAT versus 25% with enzalutamide, demonstrating the activity of BAT in this setting. With the crossover design, it was also noted that BAT followed by enzalutamide did sensitize some patients to subsequent antiandrogen therapy [23].

In these clinical reports, several patterns of response to BAT have been described [22,23]. Some patients had no response, other had a prolonged response, while others yet had an initial response followed by the development of resistance and progression. These descriptions are consistent with a substantial proportion of CRPC patients with de novo resistance to HDT and others who acquire resistance over time, after an initial response. Through understanding the mechanisms that produce de novo or acquired resistance, there is an opportunity to identify patients likely to respond to HDT or to prolong responses in those with an initial response to HDT. Moreover, these studies have identified a new clinical state: high dose testosterone resistant CRPC. To put this into context requires an understanding of how cancer adapts to a changing androgen environment, and whether these adaptations may reveal new vulnerabilities that can be exploited clinically.

### 1.3. Contributions from the Laboratory That Support Testosterone Therapy in CRPC

At initial clinical presentation, most prostate cancer patients’ tumors are dependent on androgens for growth. Unlike hormone-dependent mammary cancers, there is a paucity of reliable cell models of hormone-dependent prostate cancer. Indeed, key early studies on androgen-dependency employed the androgen-dependent Shionogi mouse mammary tumor model [24]. Many subsequent studies have employed the LNCaP human prostate line cell line [25], the growth of which, both in culture and as xenografts, is androgen-dependent. Many investigators have used LNCaP cells to explore the basis of androgen dependence and the development of androgen-independent growth. While it is beyond the scope of this paper to exhaustively review all the worthy work in this area, we acknowledge the contributions of the late Nicholas Bruchovsky [10,24], the late Shutsung Liao [8,11,12,13,14], and their collaborators in Vancouver and Chicago, respectively, whose use of preclinical models has been critical to the development of concepts that underlie our current understanding of androgens and androgen resistance in prostate cancer. Seminal studies from John Isaacs and his collaborators at Johns Hopkins have promoted the concept of testosterone administration for CRPC and developed a mechanistic understanding of how androgens can inhibit the growth of prostate cancer cells [15,16,17,18,19,26,27]. Recent studies out of the University of Washington have implicated DNA damage mediated by liganded receptors as a mechanism for the tumor-suppressive activity of supraphysiologic testosterone and suggest that patients whose tumors exhibit defects in DNA repair pathways may be particularly responsive to testosterone therapy [28,29]. The once-heretical idea that prostate cancers adapted to a low-androgen environment may be growth-suppressed by high-dose testosterone has now led to successful clinical trials [23].

### 1.4. Preclinical Studies Indicate That Prostate Cancer Carefully Titrates Androgen Signaling

In the sections that follow, we offer illustrative examples utilizing unpublished preclinical studies from our own laboratories that emphasize the observations we and others have made that have led to insights into the role of androgens in prostate cancer, and how these insights may be exploited clinically.

The LNCaP prostate cancer cell line is derived from a lymph node metastasis and is dependent on androgens for growth in cell cultures and in xenografts [25]. When switched from growth medium (10% fetal bovine serum) to low-androgen medium (5% charcoal stripped fetal bovine serum), LNCaP cell growth ceases. A careful dose response of early passage LNCaP (≈30 from derivation) to the nonmetabolizable androgen, R1881, reveals an important but underacknowledged feature: growth of LNCaP cells exhibits a biphasic dose response to androgen exposure. Growth is maximally stimulated between 10 and 30 pM R1881 to a level approaching that seen in growth medium, but falls off dramatically at higher doses (Figure 1, red). We have observed that a biphasic dose response is also seen with dihydrotestosterone instead of R1881, and in the androgen-responsive VCaP prostate cancer cell line. The implication of data such as these is that growth of LNCaP cells, while requiring androgens, is carefully balanced between insufficient and excessive androgen signaling for optimal growth.

When growth medium is supplemented with a high level of R1881 (10 nM), LNCaP cell growth is partially inhibited, and a subset of cells undergo apoptosis. Nonetheless, LNCaP cells adapt to continuous exposure to high levels of R1881 in growth medium and will resume growth. R1881-adapted cells exhibit a dose response that is right-shifted to become a classic S-shaped curve (Figure 1, blue), unlike the biphasic response of the parental line (Figure 1, red). LNCaP cells have been widely used and often passaged extensively. We have observed a propensity for LNCaP cells to exhibit some right shift in their biphasic growth response to androgen after repeated passaging. Together, such observations imply that LNCaP cells have a well-defined androgen-sensing system that regulates growth in response to the environment.

Like many investigators, we selected LNCaP cell variants as models of CRPC through long-term culturing in low-androgen medium. The CRPC LNCaP variant, P1, was derived from a parental LNCaP line (P0) that had previously been engineered to contain an androgen-responsive luciferase reporter gene. Two other independent lines (M1, M2) were selected similarly from the M0 parental line (see the Materials and Methods section). A fourth CRPC model, 4D1enz1, was derived by selection of M0 in the androgen antagonist, enzalutamide, followed by further selection in low-androgen medium (Table 1). As models, these CRPC variants reflect many of the properties of human CRPC, including overexpression of AR mRNA and protein that enables the cells to adapt to a low-androgen environment (Figure 2 and Appendix A). Notably, we did not observe expression of mRNA, nor protein, for the truncated V7 version of the AR. In addition, Table 1 lists the variant lines derived from the CRPC variants by further manipulation of their androgen environment. The characterization of these high-testosterone-resistant lines and androgen-independent lines will be addressed below.

The adaptation of CRPC LNCaP variant lines to a low-androgen environment was evidenced by their growth properties in vivo. The sensitivity of CRPC cells to androgens is highlighted in the xenograft data shown in Figure 3. The P1 CRPC model was injected into both flanks of both male and female NOD/SCID and nude mice. Tumor growth of the P1 CRPC model was slower in male mice of both strains compared to female hosts, suggesting that androgens may inhibit their growth, even though the mean testosterone levels of the males in these immunocompromised mouse models were more similar to levels of hypogonadal, not eugonadal, men [30]. Supporting the suggestion that androgens mediate growth inhibition of this CRPC tumor model was the observation that testosterone levels in female nude mice were less than one-tenth that of the males [31]. Interestingly, tumor growth was faster in nude females than in NOD/SCID females, though in both, tumor growth was faster than in the males. Thus, LNCaP variants selected in vitro for growth in a low-androgen environment were growth-inhibited even by the relatively low level of androgens present in intact male immunocompromised mice, predicting the potential utility of testosterone supplementation in men whose disease has progressed in the face of ongoing androgen suppression.

The response of the P1 CRPC variant to androgen supplementation was evaluated further in xenograft studies. In the experiment shown in Figure 4, castrate-resistant P1 cells were injected into one flank of female nude mice. Tumors were permitted to grow to a readily palpable volume, at which time mice with tumors of a similar volume were paired; one received a testosterone pellet implant and the other remained untreated. The tumors in untreated control mice continued to grow, whereas in the T-treated animals, all tumors ceased growth or regressed. In three of six mice, tumors regressed to the point of no longer being palpable. The tumors of the remaining three animals regressed somewhat or showed no progression for almost 2 months. One control mouse received a testosterone implant after a large tumor had formed (arrow, Figure 4). The tumor regressed by 75% over the next 6 weeks. The tumor from the one testosterone-treated mouse that exhibited little or no regression eventually relapsed, and the tumor began to grow despite ongoing testosterone treatment. Thus, cell-line-derived tumors exhibited heterogeneity in their response to testosterone, as has been seen in the human prostate cancer trials employing BAT.

### 1.5. Heterogeneity of AR Expression Accompanies the Failure of Manipulation of Androgen Status

With the exception of the relapsed tumor in the experiment above, there was insufficient tumor to analyze in the testosterone-implant cohort. Where palpable, the tissue appeared to consist predominantly of stromal elements and fibrous extracellular material. The relapsed tumor was still relatively modest in size, having been one of the pair with the smallest tumor volumes at the initiation of treatment. Analysis of the relapsed tumor by hematoxylin and eosin staining and immunohistochemistry (IHC) for AR and Ki67 revealed some notable features (Figure 5, treated versus untreated). Compared to a tumor from its paired control, the relapsed tumor volume was due in part to fibrotic and necrotic components devoid of tumor cells. AR expression in the relapsed tumor was lower than in the untreated tumor with some cells or regions expressing little or no receptor. Unlike the control tumor, the testosterone-treated tumor exhibited low levels of DNA replication in much of the tumor. Thus, the relapsed tumor appeared to still be in the process of adapting to high-dose testosterone and was not yet fully resistant. Similar experiments were performed in castrated male nude mice and yielded similar results, with the exception that no relapses were seen following testosterone treatment. Earlier studies observed a higher frequency of adaptation to treatment, as evidenced by tumor progression in some animals in 5 to 6 weeks [8,12,17]. This may be due to the use of different host strains or different properties of the CRPC cell lines used, and likely reflects that these models, in the same manner as human disease, can show heterogeneous characteristics, while the overall tumor behavior with respect to androgen signaling and modulation of the androgen environment faithfully reproduces human disease. This offers the opportunity to exploit the flexibility of preclinical models to understand the mechanisms underlying the response to testosterone and the development of resistance to testosterone-based therapy.

To verify the clinical relevance of AR overexpression in CRPC cell models, we assessed AR expression by IHC in 12 tumor samples from men with progressive disease in the face of ADT. In 10 of 12 metastases, the tumors overexpressed AR to different degrees. Figure 6 shows an example of moderate overexpression (patient 1) compared to another with little to no expression of AR (patient 2). Substantial heterogeneity of AR expression from cell to cell was observed in the overexpressing tumor. The heterogeneity is most easily seen in the bottom row, where the indicated portions of the field have been magnified. The importance of heterogeneity of AR expression to the process of tumor adaptation to the androgen environment is addressed further below.

### 1.6. Models of High Testosterone Resistance

Clinical trials of supraphysiologic testosterone therapy have demonstrated a new clinical state: high-testosterone resistance. Xenograft data predicted that tumors can adapt to a high testosterone environment just as they can adapt to a low-androgen environment. However, in preclinical models, treatment of CRPC xenografts differed from the experience with humans thus far, in that very long regression (relative to life span) with no sign of recurrent tumor can be seen [8] (Figure 4). This makes the study of properties of high-testosterone resistance difficult, requiring a long treatment time course to generate tumors in a fraction of mice. We asked whether selection of the testosterone-resistant lines from CRPC lines in culture rather than in vivo might serve as an appropriate and time-efficient approach for model development. Four independent CRPC variant LNCaP lines were further selected for growth in vitro by supplementing the low-androgen medium with 1 nM R1881. While growth was initially severely suppressed, the cells survived and eventually adapted to the high androgen levels in the medium over time and repeated passages until a stable phenotype was achieved that exhibited robust growth in these conditions. These lines were named P1R1/R, M1R1/R, M2R1/R, and 4D1enz1/R (Table 1). These variants all expressed diminished levels of AR (Figure 2C and Appendix A). Examination of AR expression by IHC revealed that these high-testosterone-resistant lines had not simply reverted to a “wild-type” AR phenotype. The variants, models of high-testosterone resistance, exhibited more heterogeneity of AR expression than seen in the original parental lines or the immediate progenitor, the CRPC variant. This can be seen in Figure 2C for the P series of variants and in the Appendix A for others. In the high-testosterone-resistant P1R1/R variant, there were scattered cells that did not appear to express immunoreactive AR at all, while some continued moderate to robust expression of AR. These data supported the clinical observation that there is a limit to the efficacy of repeated reversal of the androgen environment.

To test this hypothesis, we subjected the four high-testosterone-resistant lines to a second round of withdrawal of androgens in the culture medium. Unlike the case with the initial selection of CRPC lines, the second round of selection in low-androgen medium resulted in a rather transient and partial level of growth suppression as the cells adapted readily to the low-androgen environment. These lines were named P1R1/R *, M1R1/R *, M2R1/R *, and 4D1enz1/R * (Table 1). We again assessed receptor expression by IHC (Figure 2C and Appendix A). This revealed that the P1R1/R * cell lines had acquired additional heterogeneity with respect to AR. Significant numbers of cells did not express detectable AR, and many others expressed only moderate levels of AR, unlike the initial CRPC variants in low-androgen medium. At this stage, these variants grew in either high-androgen or low-androgen medium and represent a model of androgen-independence.

To substantiate the in vitro data suggesting the capacity of high-testosterone-resistant P1R1/R cells to respond to a further round of androgen manipulation was limited, P1R1/R cells were injected into the flanks of castrated male mice with implanted testosterone pellets. Tumors were allowed to develop; at which time the testosterone pellets were removed from half of the mice. Androgen withdrawal had only a modest, transient effect on tumor growth, supporting the in vitro observations that these cells were relatively resistant to further rounds of manipulation of their androgen environment. Histochemical analysis (Figure 7) did show evidence for fibrotic tissue within the testosterone-withdrawn tumors, but DNA replication was robust in both the testosterone-replete controls and the testosterone-withdrawn tumors. In this setting, testosterone withdrawal no longer elicited high AR expression. Receptor expression was heterogeneous with some cells and some areas of the tumor expressing little or no receptors. These data indicated that the tumor growth had become largely independent from the androgen environment, and that there was limited capacity to manipulate the androgen environment before growth truly became independent of androgen status.

## 2. Materials and Methods

### 2.1. Cell Lines

Prostate cancer cell line LNCaP was obtained from the University of Colorado Cancer Center (UCCC, Aurora, CO, USA) Tissue Culture Shared Resource and were authenticated by single tandem repeat analysis. These cells were the ultimate parent of all the variants described in Table 1. The experiment presented in Figure 1 employed early-passage LNCaP cells obtained directly from the originator [25], and were authenticated as described [32,33]. LNCaP cells were routinely grown in RPMI1640 medium (GIBCO, Waltham, MA, USA) supplemented with 10% fetal bovine serum.

### 2.2. Selection of LNCaP Variants

The P0 variant was derived by transducing LNCaP cells with an ARE-luciferase reporter lentivirus (Cignal Lenti AR reporter, Qiagen CSL-8019L (Germantown, MD, USA) and selection for puromycin resistance. The M0 variant was selected independently in the same way as P0. The enzalutamide-resistant 4D1 line was derived by repeated passaging of LNCaP cells in increasing concentrations of enzalutamide between 1.25 and 2.5 μM for 3 months.

The P0, M0, and 4D1 variants were then used to generate models of CRPC by culturing cells in low-androgen medium (RPMI1640 supplemented with 5% charcoal-stripped fetal bovine serum). Growth slowed, and the cells acquired a dendritic morphology. After 8–10 months of continuous culture in low-androgen medium and passaging as necessary, the cells exhibited a stable phenotype featuring robust growth in low-androgen medium, increased AR expression, and growth inhibition and cell death in response to treatment with 1 nM R1881. These variants were named P1, M1, M2, and 4D1enz1. M1 and M2 were independently derived from M0. While the overall behavior of the four CRPC models with respect to androgen signaling was similar, there were clear differences between them in response to certain drug treatments, reflecting the heterogeneity seen in human CRPC.

These CRPC variants were subjected to selection by growth in low-androgen medium supplemented with 1 nm R1881. This initially inhibited growth and stimulated apoptosis, but the cells eventually adapted, and after repeated passaging in these conditions, reacquired robust growth. These lines represented models of high-testosterone resistance (HTR) and were named P1R1/R, M1R1/R, M2R1/R, and 4D1enz1/R. They featured greatly reduced levels of AR expression compared to their CRPC progenitors.

Finally, to test whether repeated rounds of manipulating the androgen environment continued to inhibit growth, the testosterone-resistant lines were cultured in low-androgen medium without R1881 supplementation. At this point, growth inhibition was far less pronounced than in the initial round of selection in low-androgen medium. Cells adapted within a few passages and resumed robust growth in low-androgen medium. Further manipulation of the androgen environment of these lines had only minimal transient effects on growth and were, therefore, defined as androgen-independent (AI) for growth. See Table 1 for a listing of the entire set of variants and their phenotypes.

### 2.3. Western Blots

Cell pellets were collected and resuspended in a lysis solution. Twenty micrograms of protein were separated by polyacrylamide gel electrophoresis and transferred to a nitrocellulose membrane (Invitrogen, Waltham, MA, USA). The membrane was blocked and incubated overnight with the primary antibody at 4 °C. The AR N-20 antibody sc-816 was purchased from Santa Cruz Biotechnology (Dallas, TX, USA). The GAPDH antibody, 1574S, was purchased from Cell Signaling Technology. Blots were washed 3 times in PBST followed by 1 hour of incubation with secondary antibodies conjugated to horseradish peroxidase (Invitrogen). GAPDH was used as a loading control. Band signals were visualized with a LI-COR system (Li-COR Odyssey CL× Imaging System, (Lincoln, NE, USA).

### 2.4. Immunohistochemistry

Histological analyses were performed on formalin-fixed cells or tissues. For the latter, 5 µM sections were used. The sources of primary and secondary antibodies and methodological details for their use follows: androgen receptor (Cell Marque, #200R-15 (Rocklin, CA, USA), 1:200, secondary: (Agilent Technologies, EnvisionPlus Rabbit, Dako # K4003, Santa Clara, CA, USA); Ki67 (Thermo Scientific, #RM-9106-S1, Waltham, MA, USA) 1:400; secondary: horse anti-rabbit, (Vector Labs # MP-7401, Burlingame, CA, USA).

### 2.5. Tumor Xenograft Studies

The 4- to 6-week-old mice were purchased from Charles Rivers Laboratories (NU/NU nude strain 088, NOD/SCID strain 394). Tumor xenografts were generated by injecting cells into the flanks at 8–9 weeks. In some experiments, testosterone pellets (Innovative Research of America, NA151, 12.5mg/pellet for 90-day release) were implanted in the treated group using a 10 gauge trochar (Innovative Research of America, MP-182, Novi, MI, USA) according to the manufacturer’s instructions. All procedures were carried out under protocol 495 approved by the Institutional Animal Care and Use Committee of the University of Colorado.

## 3. Discussion

The successful clinical application of BAT [21,22,23] brings along a host of critical clinical questions. The BAT approach for high-dose testosterone delivery administers a high-dose intramuscular injection monthly. This yields transient elevation of testosterone to supraphysiologic levels, with a slow decline to the normal range before the subsequent dose is given. Is this regimen of transiently elevating testosterone the best strategy, or would deeper and/or longer responses be achieved by maintaining high testosterone levels via a transdermal approach; for example, until evidence of progression is seen? There is evidence from models that a transient exposure to androgens is able to cause a sustained growth inhibition [34]. Do transient testosterone spikes vis-à-vis a maintained high-testosterone regimen render the cancer more or less likely to adapt to any further manipulation of the androgen environment? Preclinical models have the potential to address some of these questions and optimize testosterone therapy. For example, preclinical models suggest that sudden changes in the androgen environment provide superior growth inhibition. Therefore, upon cessation of testosterone therapy, would it be preferable to impose a sudden block of AR signaling by administering a strong androgen antagonist? The University of Colorado is sponsoring an early phase trial, “Square Wave Testosterone Therapy in Castration Resistant Prostate Cancer”, examining the use of high-dose transdermal testosterone delivery (ClinicalTrials.gov Identifier: NCT03734653). In this pilot study, the feasibility of attaining, and then sustaining, adequate testosterone levels with this approach is being tested, alternating with enzalutamide to provide a “square wave” testosterone dosing, compared to a BAT approach. This trial is open currently, although results are not yet available.

A better understanding of the impact of androgen levels in prostate cancer may have many clinical applications. For example, patients with advanced prostate cancer have traditionally been started on gonadotropin-releasing hormone (GnRH) agonist therapy (e.g., leuprolide). This approach does produce a transient testosterone spike or flare in the first few weeks, before a decline in the testosterone levels. The sensitivity of prostate cancer cells to manipulation of their androgen environment suggests that the testosterone flare associated with initial implementation of GnRH agonist therapy could potentially enhance the response to the subsequent reduction in testosterone levels. Since AR levels rapidly fall in response to elevated testosterone, the testosterone flare has the potential to prime cancer cells for a strong response when testosterone levels then decline to castrate levels. On the other hand, since part of the adaptation of prostate cancer to manipulation of the androgen environment is to increase heterogeneity of AR expression, it may be that any potential benefit of the testosterone flare is counterbalanced by the consequences of this first round of adaptation to the spike in testosterone levels that accompany the initial response to GnRH agonists.

Despite the questions that remain, HDT is a strategy that may have great benefit to a subset of men who have progressed to castrate-resistant prostate cancer and is worthy of additional investigation. It is imperative that predictive markers be found to best identify patients with castrate-resistant disease that are likely to respond to HDT. Human CRPC often exhibits elevated AR expression much like the variants developed from LNCaP cells, but tumors from other individuals may express modest or even no AR expression, and therefore are unlikely to achieve any sustained benefit from HDT.

The new clinical state of resistance to testosterone therapy means that models to understand the adaptations acquired by testosterone-resistant prostate cancer must be exploited to optimize the benefits of testosterone therapy. One arena in which the experimental advantages of preclinical models may be exploited is to use an integrated omics approach to identify pathways that are associated with adaptation to testosterone resistance. These adaptations could reveal vulnerabilities that are amenable to targeting with existing drugs. Current studies in our laboratories have identified candidate strategies that are being evaluated at this time.

## 4. Conclusions

The work of many investigators has highlighted the balancing act that prostate cancer must maintain with respect to androgen signaling. Preclinical models have been instrumental in overcoming the mindset that androgens cannot be included in the armamentarium of prostate cancer therapy. High-dose testosterone has now been shown to be of potential therapeutic value in a subset of men whose cancer has progressed while on ADT. Preclinical models offer the prospect of identifying the mechanisms that underlie adaptation of tumors to HDT. Understanding these mechanisms, it is hoped, will lead to improvements in HDT and patient selection, as well as new approaches to overcome resistance.

## Figures and Tables

**Figure 1 life-11-00884-f001:**
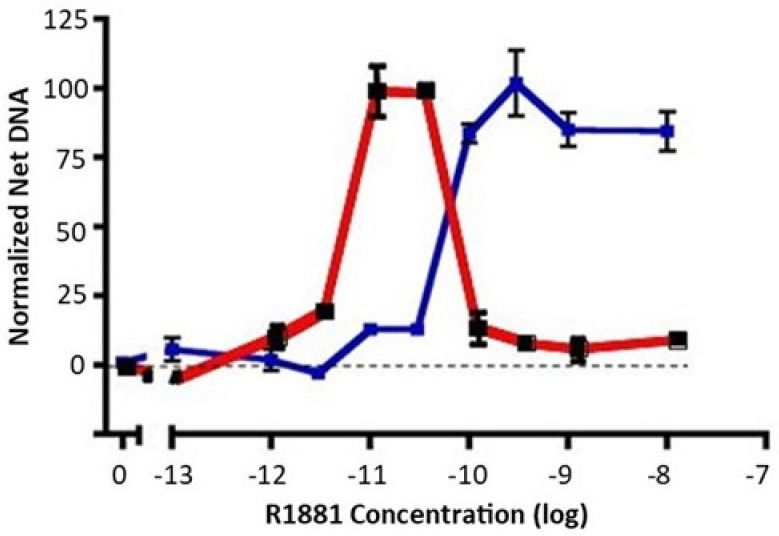
Dose response of early-passage LNCaP cells and high-testosterone-resistant LNCAP cells to R1881. Net growth of early-passage LNCaP cells was determined as described in the Materials and Methods section. The results of two experiments are shown. The dose response of early-passage LNCaP cells to R1881 is depicted in red. The blue line depicts the right-shifted dose response of a high-testosterone-resistant variant. To compare experiments, the net growth at optimal R1881 levels for that cell line was normalized to 100. Maximal growth was similar for the two lines, about an 8-fold increase in DNA content. Each condition was assessed in 6 separate wells; vertical bars represent one standard error.

**Figure 2 life-11-00884-f002:**
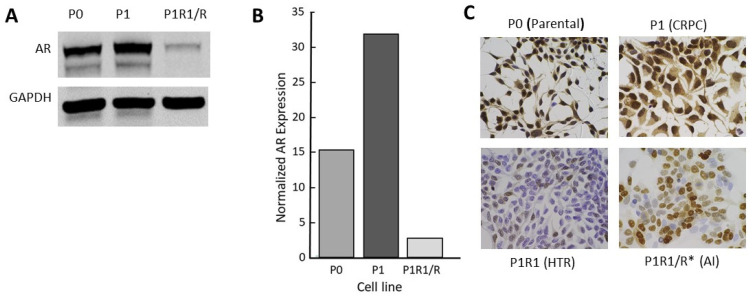
Characterization of the androgen receptor in a CRPC LNCaP model: (**A**) Western blot for full length AR; (**B**) quantitation of data from Western blots; (**C**) AR immunohistochemistry (IHC) for P series variants: P0 (parental), P1 (CRPC), P1R1/R (HTR), and P1R1/R * (AI). IHC for the other variant series is presented in Appendix A. The whole western blot image is provided in Appendix A.

**Figure 3 life-11-00884-f003:**
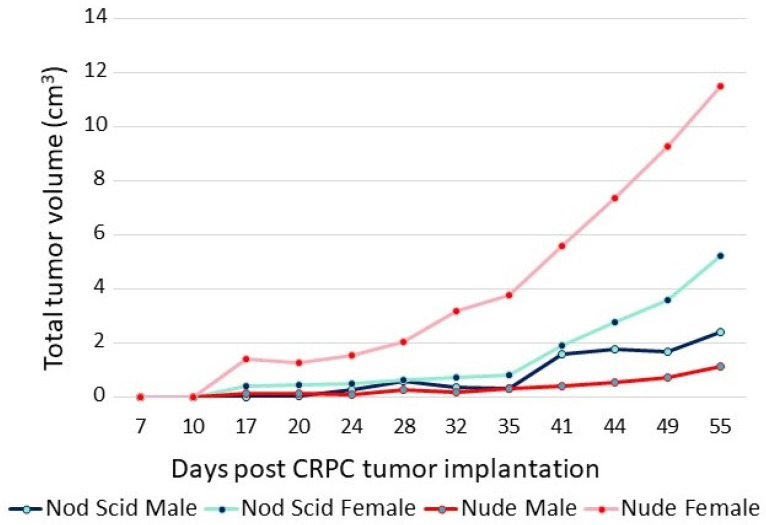
Growth of P1 (CRPC LNCaP variant) in male and female nude and NOD/SCID mice. A total of 5 × 10^6^ cells were injected into each flank of 5 mice from each group. The sum of the volumes of all tumors within a group over time is displayed.

**Figure 4 life-11-00884-f004:**
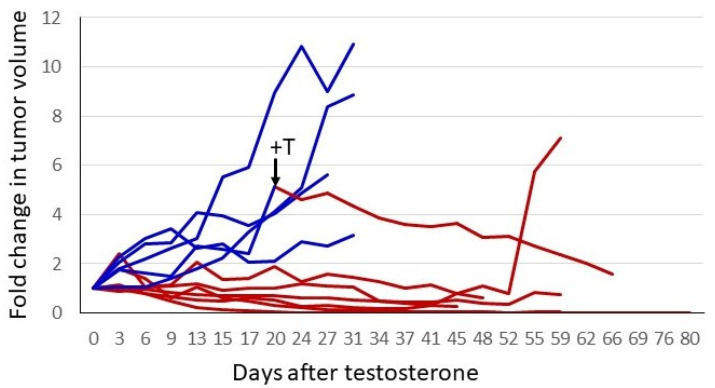
Testosterone treatment of established CRPC tumors. A total of 2 × 10^6^ P1 cells (CRPC LNCaP variant) were injected into the right flank of 12 female nude mice. When tumors reached a readily palpable volume (between 100 and 350 mm^3^), animals with similar tumor volumes were paired. One of each pair had a testosterone pellet implanted. To normalize for different starting tumor volumes, the fold-increase was plotted. Blue represents tumor growth over time in control animals, and red depicts animals that received testosterone. One untreated animal had a testosterone pellet implanted after the tumor had grown to a volume that would soon mandate euthanasia. This is indicated by the arrow and +T in the figure. The switch in treatment condition is also indicated by the change in the line plot from blue to red.

**Figure 5 life-11-00884-f005:**
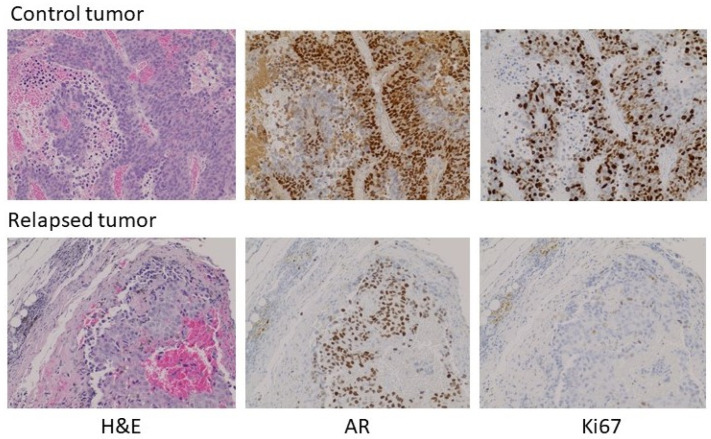
Histochemical analysis of a control tumor (top) and a relapsed testosterone-treated tumor (bottom) from the experiment in Figure 4. Left to right: H&E stain, IHC for AR, IHC for Ki67. 100× magnification. A larger field view can be seen in Appendix A.

**Figure 6 life-11-00884-f006:**
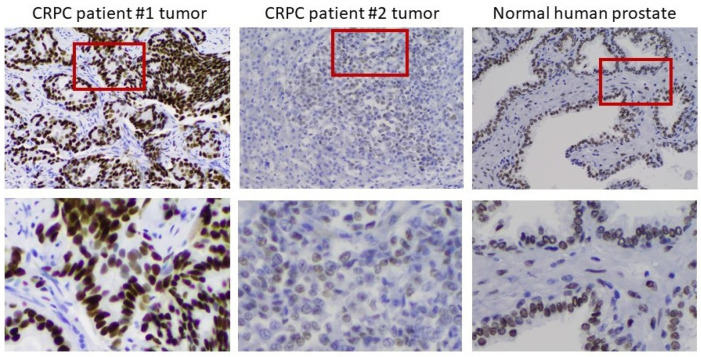
Heterogeneity of AR expression in metastases from men who have progressed on ADT. Left to right: IHC on a moderately AR overexpressing tumor, an AR-deficient tumor, and normal prostate. The bottom row shows a magnification of the indicated portion of the field above to better demonstrate the heterogeneity of AR expression in the tumors.

**Figure 7 life-11-00884-f007:**
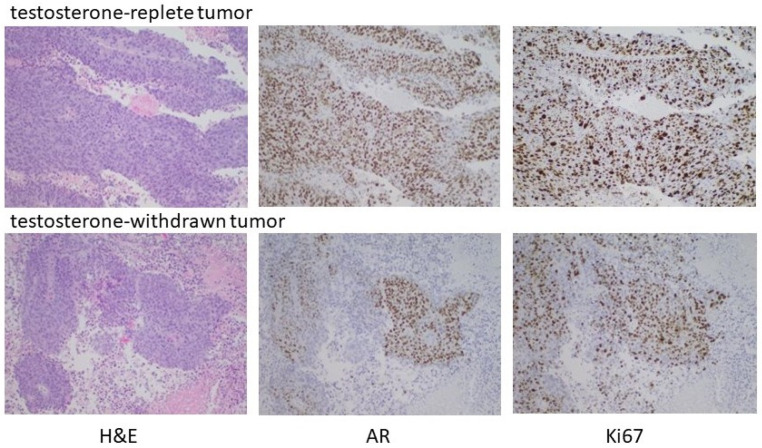
Testosterone withdrawal of established HTR tumors. Male nude mice were castrated at 7 weeks. Testosterone pellets were implanted into 12 castrated male mice one week after surgery. One week later, 2 × 10^6^ P1R1/R cells (HTR LNCaP variant) were injected into the right flank. After the development of readily palpable tumors, the testosterone pellets were removed from half of the mice and tumor development assessed. As indicated in the text, testosterone withdrawal had, at best, partial and transient regression in some animals. Histochemical analyses of representative tumors are shown.

**Table 1 life-11-00884-t001:** The four different series of variant cell lines used in studies of the effect of manipulation of the androgen environment on cell growth.

Cell Line	Parent	Selection	Phenotype
P0	LNCaP	ARE-luc	wild type
P1	P0	low androgen	CRPC
P1R1/R	P1	high androgen	HTR
P1R1/R *	P1R1/R	low androgen	AI
M0	LNCaP	ARE-luc	wild type
M1	M0	low androgen	CRPC
M1R1/R	M1	high androgen	HTR
M1R1/R *	M1R1/R	low androgen	AI
M2	M0	low androgen	CRPC
M2R1/R	M2	high androgen	HTR
M2R1/R *	M2R1/R	low androgen	AI
4D1	LNCaP	enzalutamide	enz-resistant
4D1enz1	4D1	low androgen	CRPC
4D1enz1/R	4D1enz1	high androgen	HTR
4D1enz1/R *	4D1enz1/R	low androgen	AI

ARE-luc means that an androgen response element driven luciferase reporter gene was stably transduced into the parent LNCaP cells to generate the P0 and M0 variants. CRPC, model of castrate-resistant prostate cancer; HTR, model of high testosterone-resistant prostate cancer; AI, model whose growth was independent of or transiently affected by manipulation of the androgen environment. “*” indicates variant lines selected from HTR variants by culture in low androgen medium containing 5% charcoal-stripped serum.

## Data Availability

This manuscript comprises a review of the current clinical and basic science understanding of high testosterone with prostate cancer. Some original data and figures are included for illustrative purposes. Please contact the authors if there are questions about these primary data.

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
