# Peer review of "Titration of Androgen Signaling: How Basic Studies Have Informed Clinical Trials Using High-Dose Testosterone Therapy in Castrate-Resistant Prostate Cancer"

_life, 2021, doi:10.3390/life11090884_

Round 1

Reviewer 1 Report

The manuscript is really interesting and provides a lot of information, however you should have in mind that it should be readable for all: students, patients as well as doctors and specialist in the field. Your manuscript is written for the specialist- it would be really hard to maintain the focus for students and not-specialist. Thus, I would recommend to provide a short summarize information before you present the data to make it clear for the reader what you would like to present. The same story for basic information: it is really nice to provide a one or two sentences to provide a basic information and then provide the more sophisticated.

Moreover, parts of the manuscript are completely different written eg. “androgens: friend and foe” and “preclinical studies indicate that prostate…”- the type of heading is different, the style of the presentation of data also. The manuscript should be clear and concise in all sections. Moreover, the manuscript might be divided for the review part and experimental part, and I am not so sure about this kind of data presentation in that form, cause as mentioned before the style of presentation is completely different and should be unified.

  1. Please provide a biography as a separate file not a part of manuscript on the title page
  2. Put more focus for editorial preparation e.g. line 65- different font in the abstract, line 114 point is missing
  3. Although it is review manuscript, a short introduction should be provided concerning the importance of the topic
  4. I am not sure if “titration” is the best word, maybe you use it, but the title should be obvious for all readers, thus I would recommend to change it to the more common
  5. Line 84, CYP17- provide a full name for the first time; line 92, usually we used a short for prostate cancer (PC/PCa), line 202 a point in the heading
  6. My suggestion as a reader: Line 116- the heading usually helps the reader to understand and organize the manuscript, thus “Androgens: friend and foe?” won’t help the reader to organize the knowledge. It is nice for the title of the manuscript, not the heading.
  7. You present your own results, but was it published before or not, cause there is no information about it, and should be clearly stated if so.
  8. Figure 2- a fonts in the figure should be adjusted to the requirements of the journal.

Reviewer 2 Report

In the current review article, authors did an elegant job in summarizing and discussing the important and milestone basic biology studies that have contributed to the advancement of clinical trials in high-dose testosterone (HDT) therapy treating castration-resistant prostate cancer (CRPC). Authors thoroughly reviewed the development of prostate cancer (PCa) therapy, and PCa therapies with androgen, such as HDT and Bipolar Androgen Therapy (BAT). Authors went on discussed the important molecular studies that provided the possible mechanistic basis of the HDT and BAT and further discussed potential pitfalls of HDT, and suggested possible mechanisms and improvement in future HDT clinical trial design. I personally find it is very informative and enjoyable to read. This review article will benefit the field greatly.

Author Response

We truly appreciate the reviewer's laudatory comments. As no suggestions for changes were made by this reviewer, we've no further comment other than changes to the manuscript were made in respone to the other reviewer.